# Augmented manipulation ability in humans with six-fingered hands

C. Mehring [1,2,9], M. Akselrod[3,4,9], L. Bashford [1], M. Mace[5], H. Choi[1], M. Blüher[1], A.-S. Buschhoff[1], T. Pistohl[1], R. Salomon[6], A. Cheah[7], O. Blanke[8], A. Serino[3] & E. Burdet [5]

Neurotechnology attempts to develop supernumerary limbs, but can the human brain deal with the complexity to control an extra limb and yield advantages from it? Here, we analyzed the neuromechanics and manipulation abilities of two polydactyly subjects who each possess six fingers on their hands. Anatomical MRI of the supernumerary finger (SF) revealed that it is actuated by extra muscles and nerves, and fMRI identified a distinct cortical representation of the SF. In both subjects, the SF was able to move independently from the other fingers. Polydactyly subjects were able to coordinate the SF with their other fingers for more complex movements than five fingered subjects, and so carry out with only one hand tasks normally requiring two hands. These results demonstrate that a body with significantly more degrees-of-freedom can be controlled by the human nervous system without causing motor deficits or impairments and can instead provide superior manipulation abilities.

[1] Bernstein Center Freiburg, University of Freiburg, Freiburg im Breisgau 79104, Germany. [2] Faculty of Biology, University of Freiburg, Freiburg im Breisgau 79104, Germany. [3] Department of Clinical Neurosciences, University Hospital Lausanne (CHUV), Lausanne 1005, Switzerland. [4] Cognition, Motion and Neuroscience Unit, Minded Programme, Fondazione Istituto Italiano di Tecnologia, Genova 16152, Italy. [5] Department of Bioengineering, Imperial College of Science, Technology and Medicine, London SW7 2AZ, UK. [6] Gonda Brain Research Center, Bar Ilan University, Ramat Gan 5290002, Israel. [7] Department of Hand & Reconstruction Microsurgery, National University Hospital, Singapore 119228, Singapore. [8] Center for Neuroprosthetics, Swiss Federal Institute of Technology of Lausanne (EPFL), Geneva 1202, Switzerland. [9] These authors contributed equally: C. Mehring, M. Akselrod. Correspondence and requests for materials should be addressed to C.M. (email: carsten.mehring@biologie.uni-freiburg.de) or to E.B. (email: e.burdet@imperial.ac.uk)

Additional artificial limbs that are seamlessly controlled concurrently to the natural limbs, and can assist actions with little cognitive effort, are a popular idea in science fiction and art. Inspired by this vision, engineers have undertaken to design wearable robotic limbs[1,2] and recent neuroscience studies have aimed at developing ways to interface such limbs with the nervous system[3–6]. However, is the human brain able to control a body with additional degrees-of-freedom (dof), as the range of possible movements increases exponentially with every dof (see Supplementary Note), and could this enhance functional abilities? Furthermore, how can the nervous system represent an extra limb and its relation to other limbs? The challenging, massive reorganization of neural representation required for individuals with abnormal body structure is illustrated through phantom limbs experienced by amputees[7,8]. While the physiological consequences of a missing limb have been studied, none of the corresponding fundamental issues of movement augmentation have yet been examined in the literature.

Here we address these issues by analyzing for the first time the neuromechanics and manipulation abilities of the right hand in two polydactyly subjects (17-year-old subject P1 and his 52-year-old mother, subject P2), who both have six anatomically fully developed fingers on the two hands (Fig. 1a, Supplementary Fig. 1). Polydactyly, the congenital physical anomaly of hands with more than five fingers is not rare in humans, with an incidence of around 0.2%[9] and archeology has demonstrated the presence of polydactyly individuals already in the mesoamerican civilization[10]. However, supernumerary fingers are often removed at birth[11] as they are deemed not useful and are often not fully developed.

The combinatorics of polydactyly's genetics have been analyzed in seminal nineteenth century works[12] and the genes responsible for polydactyly have been identified recently[13]. However, the neuromechanics and functionality of polydactyly hands raise many questions that have never been investigated: First, is the movement of the additional finger actuated by other fingers' muscles, or does it have its own dedicated muscles and nerves? Second, how independent is the extra finger from the other fingers? Does its movement accompany the movement of common fingers, like in the little and ring fingers[14], or does it move independently from other fingers like the thumb? Third, hand movements are already among the most complex movements humans can perform, requiring a large area of the sensory and motor cortices to control them[14–16]. Therefore, how could the cortex control a hand with several additional dof? Fourth, what is the perceived body representation of polydactyly hands? Fifth, and most importantly, are the supernumerary fingers (SF) functional, and can they provide advantages in terms of additional manipulation abilities? The present case study examines these questions on two subjects with preaxial polydactyly with an SF between thumb and index finger. The results reveal dedicated muscles, nerves and neural resources that offer these polydactyly augmented manipulation abilities.

## Results

**The SF is actuated by dedicated muscles and nerves.** We first examined the anatomy of the six-fingered right hand of subject P1 (Fig. 1, Supplementary Movie 1). The most radial digit or "thumb" has two phalanges and the other five fingers three phalanges (Fig. 1a, b). The left and right hands of P1 have a similar shape, likewise for P2 (Supplementary Fig. 1). A magnetic resonance imaging (MRI) analysis revealed that the right hand of P1 had four digits with similar anatomy to the ulnar four fingers of common hands (Fig. 1b). The thumb's bones are similar in morphology to that of a normal thumb and have similar musculotendinous and neurovascular structures. However, the thumb's carpometacarpal joint is of the ball-and-socket type (Fig. 1d), with three dof including torsion, while a normal thumb's carpometacarpal joint is a saddle joint that does not allow torsion. The extra or supernumerary finger (SF) with three phalanges has a saddle joint similar to that of a normal thumb (Fig. 1e). It has two extrinsic flexor tendons as well as a normal extensor apparatus (Fig. 1c), in addition to dedicated digital nerves. Hence, this polydactyly hand is controlled by more muscles and nerves than normal five-fingered hands. Critically there are intrinsic muscles whose origin is the second metacarpal and whose insertion is to the proximal phalanx of the finger, similar to the muscles of a normal thumb and yielding a spherical range of motion (Fig. 1b, c).

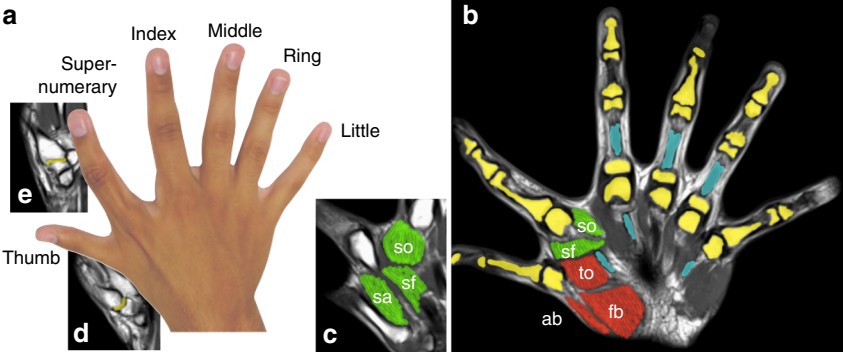

**Fig. 1** The right hand's anatomy of subject P1. A photo of the dominant right hand of one of the six-fingered subjects (**a**). The joints of the wrist (radio-ulnar, radio-carpal and mid-carpal) are similar to that of a normal five-fingered hand (**b**). Bones are in yellow, tendons in blue, muscles: so,sf,sa: supernumerary finger opponens, flexor, abductor; to: thumb opponens; ab: abductor pollicis brevis; fb: flexor pollicis brevis. The four fingers from index to little have a similar skeleton, musculotendinous attachments and nerves as the corresponding fingers of a normal hand. The thumb resembles a normal thumb, with two phalanges. However, its carpometacarpal joint to the wrist (**d**) is of ball-and-socket type, with three degrees-of-freedom (dof) including torsion, while a normal thumb will have a saddle joint that does not allow torsion. The musculotendinous and neurovascular structures resemble the thumb of a normal hand (**b**, **d**). The sixth finger or supernumerary finger has three phalanges and a saddle carpometacarpal joint (**e**). It has two extrinsic flexor tendons and a normal extensor apparatus not dissimilar to that of a tri-phalangeal digit. Interestingly, there are muscles whose origin is the second metacarpal and whose insertion is to the proximal phalanx of the finger (**b**, **c**), similar to the muscles of a normal thumb with spherical range of motion

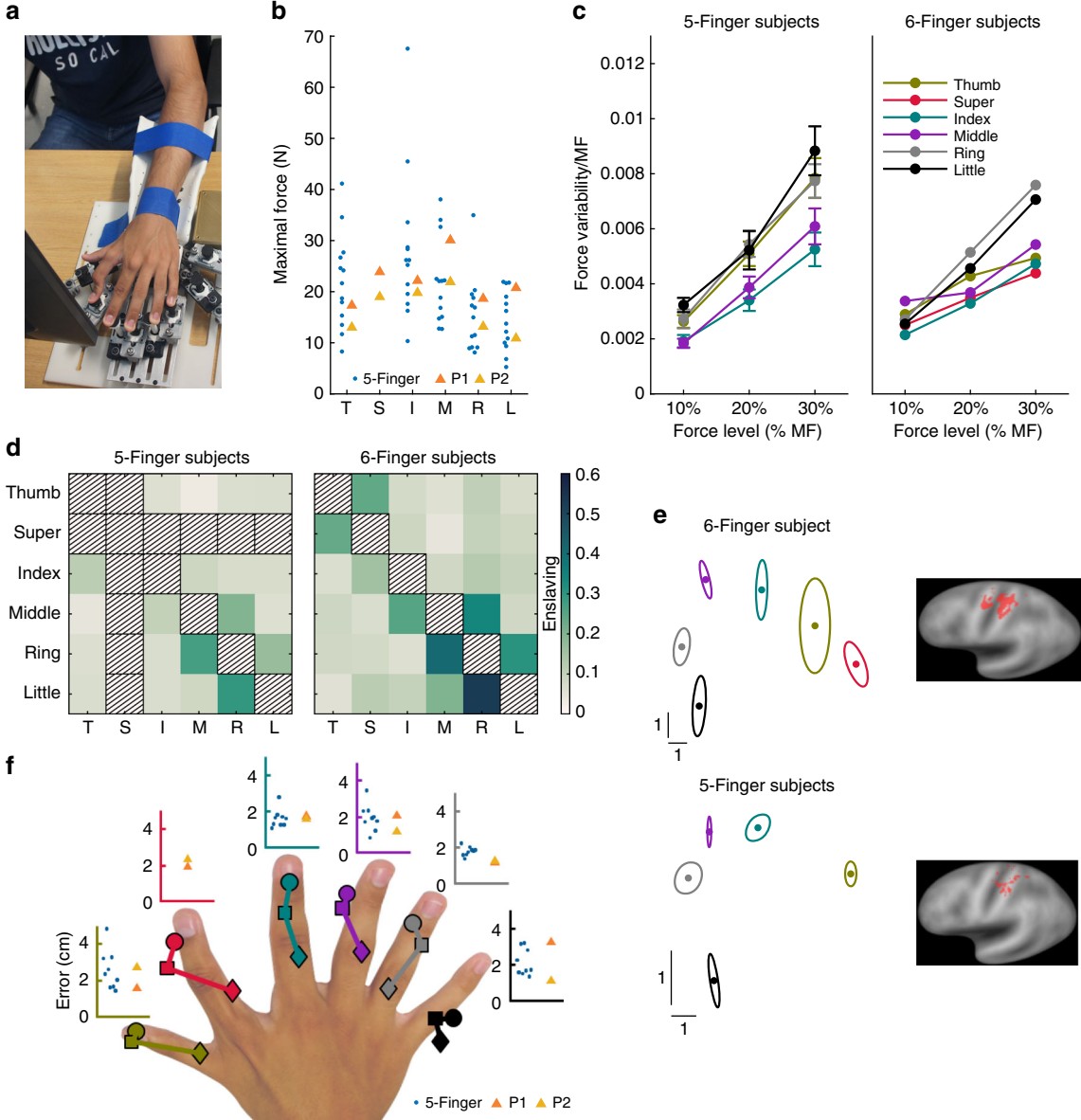

**Fig. 2** Neuromechanics of the polydactyly hand. **a** Dedicated isometric interface to investigate the force capability of each finger in individuals with five- and six-fingered hands. The interface was used for the analyses presented in subplots (**b–d**). Subjects were initially asked to exert maximal force (MF) with a single finger. In a consecutive experiment subjects were asked to control 10%, 20%, or 30% of MF during 15 s long trials. Two six- and 13 five-fingered subjects carried out these experiments. **b** MF produced by individual fingers. The MF was similar for five- and six-fingered subjects for all regular fingers. **c** Force variability (standard deviation of the force) as a function of the magnitude of the produced force expressed in percentage of the maximal force. Error bars depict SEMs across subjects. **d** Enslaving shows the forces induced in other fingers when the subject is instructed to exert the maximal force in one finger. Enslaving between finger $i$ and $j$ was computed as $e_{ij} = \frac{F_j(i)}{MF_j}$, where $F_j(i)$ is the force produced by finger $j$ when finger $i$ was instructed to produce maximal force. $MF_j$ depicts the maximal force of finger $j$. In the matrix plot the instructed finger is shown on the $y$ axis, hence, each row shows the induced force relative to the maximal force of the corresponding finger, i.e. $e_i$. **e** Two-dimensional projection (multidimensional scaling, MDS) of fMRI activation in sensorimotor cortex during individual finger movements in subject P1 and the average across nine five-fingered control subjects (left). Colors depict different fingers as in (**c**). Ellipses show the standard error of the mean. The location of the activation cluster of the supernumerary finger is separate from the activation clusters of the other fingers. Selected voxels which were used for the MDS are shown on the right. **f** Mental representation of six-fingered hands. Blindfolded subjects pointed with the index finger of one hand to a cued location (first, second knuckle or tip) on the other hand. Pointing errors were similar in the two six- and 9 five-fingered subjects and similar for the supernumerary as for other fingers

**Neuromechanics of polydactyly hands.** Using a dedicated interface to measure the force exerted by individual fingers (Fig. 2a, Supplementary Movie 2), we could then examine the fingers' biomechanical characteristics. The maximal force was similar in six- and five-fingered subjects (Fig. 2b; two-sided Wilcoxon rank sum test, $p = 0.80$, $W = 18$, 95% CIs [−14N,12N],

average across fingers six- vs. five-fingered subjects, $N_{1,2} = 2,13$) and the maximum force exerted by the SF was similar to the strongest other fingers (two-sided Wilcoxon signed-rank test, $p = 0.50$, $W = 3$, 95% CIs [0.21N,1.2N] SF vs. the average across thumb, index and middle finger, $N_{1,2} = 2,2$). Force variability increased with the force level in the SF like in other fingers of the

six-fingered hands and in the five-fingered hands (Fig. 2c)[17]. The interdependency (or "enslaving")[18] of all pairs of fingers was examined by instructing subjects to produce maximal force with each individual finger separately, while measuring the force of all fingers without providing visual feedback of the force exerted by the remaining fingers. The enslaving value was computed as the magnitude of the force exerted by a finger relative to its maximal force. The results of this analysis revealed that the SF was independent from other fingers, with only some dependence between the SF and the thumb (Fig. 2d). The other fingers had similar dependencies as in common five-fingered hands (Fig. 2d) and enslaving magnitude was highly correlated between five- and six-fingered subjects across finger pairs available in both hands (Pearson correlation coefficient, $r = 0.94$, $N = 20$). The enslaving between SF and index, little, middle, ring, little finger was not different from the enslaving between the thumb and the other fingers in five-fingered subjects (two-sided Wilcoxon rank sum test, $p = 0.30, 0.38, 0.17, 0.57$, $W = 23, 22, 25, 20$, 95% CIs $[-0.12, 0.18]$, $[-0.028, 0.052]$, $[-0.028, 0.10]$, $[-0.10, 0.13]$ for index, middle, ring and little finger, $N_{1,2} = 2, 13$). Enslaving for lower levels of force (10%, 20% and 30% of maximal force) was similar to enslaving at maximal force, in particular for 20% and 30% of maximal force (Supplementary Fig. 2). Furthermore, the independent controllability of the SF was also exemplified by our polydactyly subjects being able to do pinch grips between the SF and all other fingers (Supplementary Movie 3).

Next, we investigated the functional organization of the motor cortex in P1 using an individual finger tapping task and functional MRI at 7T high resolution[19]. In order to highlight the specific representations of each finger[14], we compared the activity patterns generated by individual finger movements (Supplementary Fig. 3). The results show that the representation of the SF in the primary sensorimotor cortices was distinct from the representations of all other fingers, including the thumb (Fig. 2e, Supplementary Fig. 4). This demonstrates that separate neural resources are used to control movements of the SF in this six-fingered subject.

**Mental representation of polydactyly hands**. To infer the mental representation of the hand in our polydactyly subjects, we asked them to indicate their perceived location of landmarks on the hand (fingertip, first and second knuckles, for each finger) by pointing with the other hand to the corresponding target on a two-dimensional graded grid placed above the hidden hand, following a tactile cue on the target. As we see in Fig. 2f, the hand representation corresponds to its anatomy, with the SF perceived correctly between the thumb and index. We found similar localization errors in the six-fingered and five-fingered subjects (two-sided Wilcoxon rank sum test, $p = 0.33$, $W = 7$, 95% CIs $[-0.89$ cm, 0.47 cm], average across fingers six- vs. five-fingered subjects, $N_{1,2} = 2, 9$, cf. ref. [20]). This morphologically correct representation of the fingers may support six-fingered manipulation.

**Supernumerary finger yields augmented manipulation abilities**. We then investigated the fingers' functionality by measuring their movement during free manipulation of selected objects with various shapes[21], as well as during common tasks (Supplementary Fig. 5, Supplementary Movies 4 and 5). An accurate motion capture system was used to record the movement of the distal and proximal phalanges of each finger. Interestingly, we found the same interdependencies between the fingers' movements (Fig. 3a for object manipulation, Supplementary Fig. 6A for common tasks) as in the previous biomechanical investigation, suggesting that the mobility and independence of the SF is not reduced during manipulation ($r = 0.88$, $N = 30$, Pearson correlation

coefficient between enslaving matrix in Fig. 2d and dependency matrix in Fig. 3a for six-fingered subjects; $r = 0.87$, $N = 30$, Pearson correlation coefficient as before but using the dependency matrix for common tasks shown in Supplementary Fig. 6A). The interdependencies between the fingers' movements was highly correlated between five-fingered and six-fingered hands for all fingers excluding the SF ($r = 0.996$, $N = 10$, Pearson correlation coefficient for object manipulation; $r = 0.995$, $N = 10$, Pearson correlation coefficient for object manipulation). The movements of the SF, like the thumb and index finger's movements, could not be reconstructed from the movements of the other fingers (Supplementary Fig. 7). Consistently, an examination of the fingers' kinematic synergies[22] revealed that movements of the six-fingered hands had a higher number of effective degrees of freedom than five-fingered hands (Fig. 3b, c, Supplementary Fig. 6B,C; two-sided Wilcoxon ranksum test, $p = 0.019$, $W = 29$, 95% CI [2.5,6.9], $N_{1,2} = 2, 13$ for object manipulation; $p = 0.044$, $W = 19$, 95% CI [1.5,8.6], $N_{1,2} = 2, 8$ for common tasks). These results were confirmed by an information theoretic analysis (Fig. 3d, Supplementary Fig. 6D) taking into account nonlinear relations. The movement of each finger was classified into one of three states {rest, flexion, extension} yielding $3^f$ ($f$: number of fingers) different movement configurations on which the joint entropy was calculated (Fig. 3d, Supplementary Fig. 6D). For finger combinations available in both kinds of subjects the entropy for six-fingered subjects was similar to five-fingered subjects (two-sided Wilcoxon rank sum test, $p = 0.17, 0.23, 0.23, 0.30, 0.38$, $W = 25, 24, 24, 23, 22$, 95% CIs in bits $[-0.011, 0.055]$, $[-0.018, 0.072]$, $[-0.11, 0.26]$, $[-0.079, 0.31]$, $[-0.18, 0.38]$, for the first five-finger combinations shown in Fig. 3d for object manipulation, $N_{1,2} = 2, 13$; two-sided Wilcoxon rank sum test, $p = 0.53, 0.27, 0.044, 0.044, 0.40$, $W = 8, 16, 19, 19, 15$, 95% CIs in bits $[-0.080, 0.11]$, $[-0.056, 0.16]$, $[0.046, 0.20]$, $[0.12, 0.47]$, $[-0.097, 0.66]$ same combinations for common tasks, $N_{1,2} = 2, 8$). However, the maximal entropy for six-fingered hand movements was substantially higher than for five-fingered movements (two-sided Wilcoxon rank sum test, $p = 0.019$, $W = 29$, 95% CI in bits [1.3,1.8], $N_{1,2} = 2, 13$ for object manipulation; two-sided Wilcoxon rank sum test, $p = 0.044$, $W = 19$, 95% CI in bits [1.2,2.1], $N_{1,2} = 2, 8$ for common tasks). Moreover, the entropy was higher than the maximum possible entropy for five fingers and close to the maximum possible entropy for six fingers showing that subjects used a rich ensemble of movement patterns. Furthermore, the SF was moved most of the time in coordination with both the thumb and index finger, rather than moving alone or with only the thumb or the index finger (Fig.3e, Supplementary Fig. 6E). Consequently, the independence of the SF could not simply be ascribed the function of replacing the thumb or index finger. Instead, six-fingered hands featured unique movement patterns involving thumb, SF and index finger. Importantly, this did not come at the expense of slower movements (Fig. 3f, Supplementary Fig. 6F): the movement speed was similar for five- and six-fingered subjects (two-sided Wilcoxon rank sum test, $p = 0.80$, $W = 18$, 95% CIs $[-1.0$ cm/s,1.5 cm/s], average across fingers in six- vs. five-fingered subjects, $N_{1,2} = 2, 13$, object manipulation; two-sided Wilcoxon rank sum test, $p = 0.27$, $W = 16$, 95% CIs $[-0.28$ cm/s,4.2 cm/s], average across fingers in six- vs. five-fingered subjects, $N_{1,2} = 2, 8$, common tasks). Taken together these results demonstrate that the movements of the six-fingered hands of our two subjects had increased complexity relative to common five-fingered hands.

To examine whether the superior functionality of six-fingered hands enabled our polydactyly subjects to carry out tasks that cannot be completed with one five-fingered hand, we designed a video game stimulating subjects to coordinate finger movements at increasing speed (Fig. 3g, Supplementary Movie 6). The video

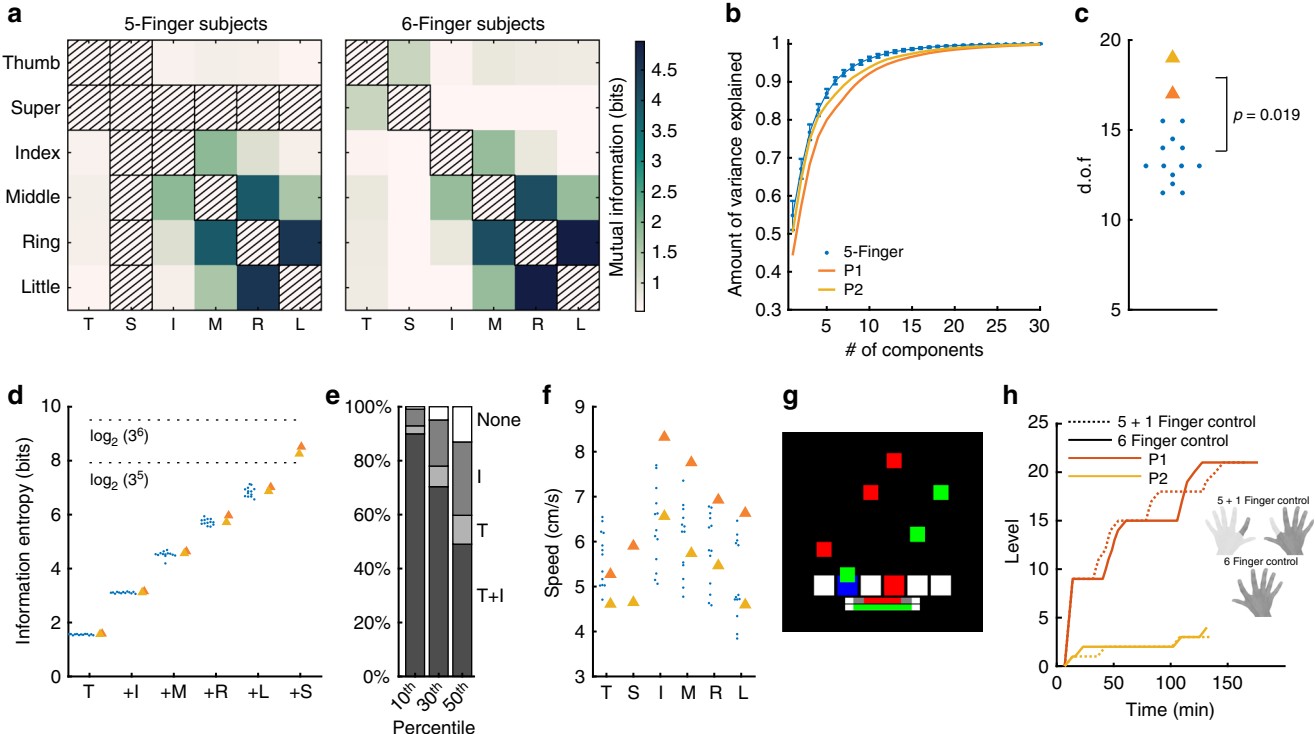

**Fig. 3** Hands with supernumerary fingers perform more complex movements. Subplots (**a**–**f**) report on analyses of hand movements recorded during manipulation of objects of various shapes. The movement task was carried out by two six- and 13 five-fingered subjects. **a** Dependency between individual fingers quantified by the mutual information between the movements of pairs of fingers, with a value of 0 indicating complete independence between fingers and positive values an increasing dependency. Note that the mutual information is symmetric, i.e. $I(X, Y) = I(Y, X)$. **b** The cumulative amount of explained variance of hand movements as a function of an increasing number of principal components. Error bars depict SDs across subjects. **c** The number of effective dof (computed using the principle components, see Methods) was higher in six-fingered than in five-fingered subjects. **d** Information entropy of the discretized movements where each finger is either resting, flexing or extending. Entropy is shown for an increasing number of fingers, starting with thumb only ("T") and successively adding one finger (index "I", middle "M", ring "R", little "L" and supernumerary "S"). Dotted lines indicate the theoretically maximum possible entropy for five- and six-fingered hands. **e** Percentage of times thumb and index finger ("T + I"), thumb only ("T"), index only ("I") were moving when the supernumerary finger moved. From left to right: different percentiles of the speed distribution were used as thresholds to separate rest from movement. **f** Median movement speed of individual fingers for five- and six-fingered subjects. Subplots (**g**, **h**) show results from the video game experiment. **g** Schematic of the task, subjects were required to press a button corresponding to the bottom white targets every time an oscillating cursor (green or red) entered the box. The target boxes flashed blue if a correctly timed press occurred and displayed as red if an incorrectly timed press occurred. Horizontal bars at the bottom of the screen displayed the fraction of correct key presses (top) and one minus the fraction of missed (bottom) key presses. **h** Subjects' learning curve for the 5 + 1 (dotted) and 6 finger (solid) control. Digits used shown in the inset in dark gray

game consisted of six boxes oscillating up and down at different frequencies on a computer screen; each time a box reached a target area at the bottom of the screen, the subject had to press a key with the corresponding finger. The aim was to keep both the fraction of missed key presses and wrongly timed key presses (i.e. when the box was not in the white area) below a specified threshold, for 2 minutes. When this objective was achieved, subjects moved on to a harder level. Across levels, the movement speed of the boxes increased, requiring temporally more precise finger movements (Fig. 3g). Subjects practiced the video game across 5 days, training on each day using either (a) the six fingers of their right hand, or (b) five fingers from the right without the SF and one finger from the left hand (Fig. 3h). The slopes of the learning curves (Fig. 3h) were not different between 5 + 1 and 6 finger control for both subjects ($p > 0.05$, Bootstrap test, see Methods). Hence, subjects achieved the same performance with six fingers from one hand as with two hands, which is how the task would be carried out by normal five-fingered hands. This demonstrates the augmented abilities for manipulation enabled by the six-fingered hand compared to a common five-fingered hand. Supplementary Movie 7 further illustrates the skill enabled by six-fingered manipulation.

## Discussion

Although polydactyly is not rare, and can be traced back at least 1000 years[10], only its genetics has, until now, been studied. This may in part be due to the belief that supernumerary fingers represent a malformation and are not useful, thus are generally removed at a young age. However, our study with two preaxial polydactyly subjects from the same family reveals fully functional supernumerary fingers (SF) and demonstrates their utility and the augmented manipulation capabilities they can provide. The observed SF has independent muscles, nerves, a dedicated cortical representation and an anatomically correct mental representation. Our polydactyly subjects can move the SF independently from the other five fingers and use it to carry out unique manipulation behaviors in particular in conjunction with the thumb and index finger.

Importantly, the possibilities offered by the SF biomechanics of our polydactyly subjects were not reduced or even modified by the neural control, as demonstrated during manipulation of various objects. The experiments demonstrated that they have no difficulty in controlling the SF in coordination with and independently from the other fingers while no movement deficits of the hand or other limbs were observed. The SF is used together

with all other fingers for more complex manipulation than in normal five-fingered individuals, at a similar speed. In particular, the highly mobile thumb and SF, both having a spherical workspace, allow these polydactyly subjects a large versatility and dexterity, yielding a higher sensorimotor ability for manipulation with one hand than in normal-bodied individuals. These superior abilities of our polydactyly subjects, which may be specific to the preaxial group of polydactyly and to the well-developed SF in our subjects, suggest to thoroughly evaluate the functionality of an SF in polydactyly infants before deciding on whether to remove it.

The present study is the first demonstration that the human nervous system is able to develop, embody and control multiple extra dof and integrate them into coordinated movements with the other limbs, without any apparent movement deficits or conflicts in the sensorimotor or mental representations. The exceptional manipulation abilities in our polydactyly subjects suggest that it may be of value to augment normal five-fingered hands with an artificial supernumerary finger. For several years, roboticists have been attempting to develop extra limbs to augment human movement abilities[1,2] and neural interfaces to control them[3–6]. The biomechanics and functionality of the polydactyly hands analyzed in this paper may be used as a blueprint for the development of robotic hands. However, it remains unclear how to implement real-time and embodied control of additional dofs yielding augmented manipulation capabilities. Polydactyly individuals with functional SF offer a unique opportunity to investigate the neural control of supernumerary limbs, analyze internal representations of the body and the limits of sensorimotor capabilities in humans.

## Methods

This section describes the series of experiments carried out by the two polydactyly subjects, P1 and P2, to investigate the neuromechanics and functions of their hands. Some experiments involved in addition a group of control subjects with five-fingered hands. The study was approved by the institutional ethics committees at the University of Freiburg, Imperial College London, EPFL and King's College London. Each subject gave informed consent prior to starting every experiment.

**MRI analysis of hand anatomy**. The underlying anatomy of the hand of subject P1 was visualized using MRI in the Department of Perinatal Imaging and Health, King's College London. T1 weighted, inversion recovery and proton density images were acquired with a 1.5 Tesla Siemens Aera system (Erlangen, DE). Images could not be acquired from subject P2 due to a metallic dental implant.

**Hand biomechanics**. A dedicated hand interface to measure the isometric force of each finger (shown in Fig. 2a) was developed at the Human Robotics group, Imperial College London, to investigate the force capability of either left or right fingers, in individuals with either five-fingered or six-fingered hands. The hand was placed horizontally on the interface as shown in Fig. 2a. Five or six of the eight 3D printed supports, each affixed to a load cell (HTC), could slide linearly to accommodate a left or right hand of any size so that the subject could comfortably exert a vertical force with the tip of each finger.

Forces across all fingers were recorded at 128 Hz. Experiments were carried out with this interface on the two polydactyly subjects as well as on a population of 13 control subjects (six females) with five-fingered hands between 25 and 35 years old. The subjects were seated in front of a table with the interface positioned on top of it so that the forearm was resting on the table in a natural position.

Initially, subjects were asked to exert the maximal possible force with a single finger. This maximal force (MF) was recorded for each finger separately starting with the thumb and ending with the little finger. Figure 2b shows the MF for five- and six-fingered subjects. Using this data, the enslaving $e_{ij}$, characterizing the dependence between fingers $i$ and $j$, was computed as

$$e_{ij} = \frac{F_j(i)}{\text{MF}_j},\qquad(1)$$

where $i$ is the finger that generates MF while $F_j(i)$ is the force produced simultaneously by finger $j$ and $\text{MF}_j$ is the maximal force of finger $j$. The enslaving for five- and six-fingered subjects are presented in Fig. 2d.

Then the subjects were asked to control 10%, 20%, or 30% of MF during 15 s long trials. Three trials were carried out at each force level, totalizing $3 \times 3 \times 5 = 45$ or $3 \times 3 \times 6 = 54$ trials per session for five- and six-fingered subjects respectively. Five-fingered subjects carried out only one session while the six-fingered subjects performed two (subject P1) or three (subject P2) sessions. The data from this

experiment were used to examine how the force variability depends on the amount of force exerted. In each trial, the force variability was computed as the standard deviation of the force across the time window [1300–1800]/128 s, which was selected so that the subjects were correctly exerting the required force during this period in almost all trials. Five trials (1 trial in a control subject, 2 trials in subject P1 and 2 trials in subject P2) were excluded from the analysis as they showed extraordinary high fluctuations of the force across time, indicating that the task was not carried out successfully on these trials. Figure 2c shows the standard deviation of the force as a function of the magnitude of the force for five- and six-fingered subjects.

We also computed the enslaving for the 10%, 20%, or 30% MF tasks (Supplementary Fig. 2). The normalization by the maximal force ($\text{MF}_j$) was replaced by 10%, 20%, or 30% of the maximal force, respectively.

**Functional MRI**. P1 and a group of nine control participants with five-fingered hands took part in the fMRI experiment. P2 was excluded due to a metallic dental implant. In a block design, participants performed a taping movement during 20 s with a single finger (20 taps per block, 1 tap per second) followed by 10 s of rest. Four blocks were performed for each finger in pseudo-randomized order (24 trials for P1 and 20 trials for controls). P1 performed two sessions, one for each hand. Controls performed only one session with the right hand. All participants were trained on the movements before entering the fMRI scanner.

Images were acquired on a short-bore head-only 7T scanner (Siemens Medical, Germany) with a 32-channel Tx/Rx rf-coil (Nova Medical, Germany). Functional images were acquired using a sinusoidal readout EPI sequence[23] and comprised 28 axial slices. Slices were placed over the central sulcus (approximately orthogonal to the central sulcus) in order to cover the primary motor cortices (voxel resolution $1.3 \times 1.3 \times 1.3$ mm$^3$; TR = 2 s, FOV = 210 mm, TE = 27 ms, flip angle = 75°, GRAPPA = 2). Anatomical images were acquired using an MP2RAGE sequence[24] in order to allow the precise localization of the precentral sulcus (see below) and for display purposes (TE = 2.63 ms, TR = 7.2 ms, TI1 = 0.9 s, TI2 = 3.2 s, TRmprage = 5 s). To aid coregistration between the functional and the anatomical images, a whole brain EPI volume was also acquired with the same inclination used in the functional runs (81 slices, voxel resolution $1.3 \times 1.3 \times 1.3$ mm$^3$, FOV = 210 mm, TE = 27 ms, flip angle = 75°, GRAPPA = 2). Subjects were scanned in supine position.

All images were analyzed using the SPM8 software (Wellcome Centre for Human Neuroimaging, London, UK). Preprocessing of fMRI data included slice timing correction, spatial realignment, smoothing (FWHM = 2 mm) and coregistration with anatomical images. Caret 5 (Van Essen Laboratory, Washington University School of Medicine) was used for surface visualization. To localize the voxels included in the analysis of activation patterns (Supplementary Fig. 3), a first GLM analysis was computed, which included one regressor per finger (6 for P1 and 5 for controls) and six rigid movements regressors. A functional mask for finger movements was defined as the active voxels in the F-contrast associated with any type of finger movement ($p < 0.05$ FWE). In addition, an anatomical mask corresponding to the sensorimotor cortex was designed using published probabilistic cytoarchitectonic maps[25–27]. The anatomical mask included the primary motor cortex M1 (Brodmann areas 4a and 4p) and the primary somatosensory cortex S1 (Brodmann areas 3a, 3b, 1 and 2). The anatomical mask was back-projected onto the native space of each participant. This led to 2190 voxels in the left hemisphere of P1 for right finger movements, 2037 voxels in the right hemisphere of P1 for left finger movements, and $343.8 \pm 417.1$ (mean ± std) voxels in the left hemisphere of controls for right finger movements (Supplementary Fig. 3).

To analyze the activation patterns within the selected voxels associated with each trial of finger movement, a second GLM analysis was computed, which included one regressor for each finger tapping trial (24 for P1 and 20 for controls) and six rigid movements regressors. Separately for each participant, the beta estimates for each tapping trial were extracted within the selected voxels (resulting in a trial × voxels matrix). These high-dimensional patterns were projected to two dimensions by classical multidimensional scaling (MDS), which finds low-dimensional projections preserving approximately the pairwise distances between the high-dimensional activation patterns[14]. As distance metric for the MDS, we used the cross-validated Mahalanobis distance[14]. For the five-fingered control group, MDS was carried out for each subject separately. As MDS projections induce an arbitrary rotation we aligned the projections of the individual subjects using Procrustes alignment[14]. Standard error ellipses shown in Fig. 2e were computed from the covariance across subjects. As the Procrustes alignment can also remove some of the true inter-subject variability[14], we used a Monte-Carlo procedure to estimate a correction and adjusted the standard error ellipses accordingly[14]. For the polydactly subject P1, we computed the covariance by bootstrapping the trials. For each bootstrap sample an MDS projection was computed. The bootstrapped MDS projections were aligned using Procrustes alignment. The standard error ellipses (Fig. 2e, Supplementary Fig. 4) were computed from the covariance across bootstrapped MDS projections, adjusted by correction factors estimated by a Monte-Carlo procedure[14].

**Finger localization task**. A finger localization task[20] was conducted to investigate the perceived hand shape of P1, P2, and of a group of nine controls. Participants

were blindfolded and their hand was placed below a structure topped by a 2D grid. They had to point on the grid with the index of the free hand towards the cued locations on the tested hand. They were required to identify three locations on each finger: the first knuckle, the second knuckle and the tip (total of 18 locations per hand for P1 and P2, and 15 locations for controls). Each location was tested six times for P1 and P2, four times for controls. The task was conducted for both hands in P1 and P2, only for the right hand in controls. The task was conducted once with tactile cueing, i.e. the target locations were touched with a plastic filament, and once with verbal cueing, i.e. the target locations were orally named. The localization error was measured for each tested location as the 2D-Euclidean distance between the reported positions on the grid and the real positions of the tested locations on the grid (Fig. 2f). Similar results were obtained with tactile and oral cueing; we only report the results from tactile cueing.

**Object manipulation and common movement tasks**. Experimental setup: The subjects were seated in front of a desk during the two tasks described below. An electromagnetic motion capture system (Polhemus Liberty 240/16-16) was used to record the hand and finger movements during the object manipulation and the common movement tasks (see Supplementary Fig. 5A). The hands were kept at 0.6 m distance from the main Polhemus system to maintain the recording noise below 0.005 mm. In total, 12 respectively 14 sensors were attached to the hand and fingers of five- or six-fingered subjects using medical tape. Every sensor measured three Cartesian coordinates for the position and three angles for the orientation relative to the main station. Each sensor was connected to the Polhemus system by plastic insulated aluminum wires. Two large sensors ($9 \times 11 \times 6$ mm$^3$ at maximum positions, 9.1 g) were placed on the skin on top of the middle and thumb metacarpal bones. The others were small sensors (spherical, 17.3 mm length, 1.8 mm outer diameter, <1 g) which were placed at the distal and proximal phalanges of each finger. Measurements were recorded at 120 Hz.

Object manipulation task: The two polydactyly subjects and 13 control subjects with five-fingered hands (six females, mean age 24.8 with standard deviation 2.0) participated in an object manipulation task. The experimental procedure for the object manipulation task was adapted from ref. [21]. We chose 50 objects with different shapes, sizes, textures and materials (see Supplementary Fig. 5B). These objects were without metal or paramagnetic materials so as to not interfere with the Polhemus measurement based on magnetic fields. The subjects were blindfolded and were given the objects one by one. They had to explore an object with one hand, and guess what it is (see Supplementary Movie 4). Each object was explored for 30 s. When an object was recognized earlier than 30 s, the subject was asked to explore special features of this object such as tips, edges etc.

Common movement tasks: The two polydactyly subjects and 8 of the 13 subjects with five-fingered hands who carried out the object manipulation task (five females, mean age 24.3 with standard deviation 2.0) also performed four common movement tasks (see also Supplementary Movie 5). Tying shoe laces: The end of two shoe laces were fixed on a table and the subjects were required to tie the laces with two hands. Flipping book pages: The subjects were given a book and had to flip pages using one hand only. Napkin folding: The subjects received a paper napkin and had to fold it into a specific shape (as used in restaurants) and in a specific sequence using both hands. Rolling a towel: Subjects were given a towel and asked to roll it into cylinders using both hands. Five minutes of movement per task was recorded during which subjects were asked to repeat the task as often as they wanted.

Data analysis: The position of every small sensor relative to the large sensor on the middle of the metacarpal bones was used for further analysis. Raw positional measurements were smoothed with a Savitzky-Golay filter (third order, length 41 sample points equivalent to 341.67 ms). Movement velocities were computed from raw positional measurements with a first derivative Savitzky-Golay filter (third order, length 41 sample points equivalent to 341.67 ms).

Analysis of finger (in) dependence: To assess the (in)dependence of finger movements we estimated the mutual information between the movements of different fingers. The mutual information between two continuous stochastic signals $X$ and $Y$ is defined as:

$$I(X, Y) = \int_X \int_Y p(x, y) \log_2 \left[ \frac{p(x,y)}{p(x)p(y)} \right] dx dy , \qquad (2)$$

where $p(x, y)$ is the joint probability density function of $X$ and $Y$, $p(x)$ and $p(y)$ are the marginal probability density functions of $X$ and $Y$. Note that the mutual information is symmetric, i.e. $I(X, Y) = I(Y, X)$. In case of multivariate Gaussian density functions Eq. (2) simplifies to

$$I(X, Y) = \frac{1}{2} \log_2 \left[ \frac{\det(\sigma_X) \det(\sigma_Y)}{\det(\sigma_{XY})} \right] , \qquad (3)$$

where $\sigma_X$, $\sigma_Y$ are the covariance matrices of the marginal densities $X$ and $Y$ and $\sigma_{XY}$ is the covariance matrix of the joint density. A more intuitive understanding of the mutual information can be gained for univariate normal signals $X$ and $Y$ for which Eq. (3) further simplifies to

$$I(X, Y) = \log_2 \sqrt{\frac{1}{1 - r(X,Y)^2}} , \qquad (4)$$

where $r(X, Y)$ is the Pearson correlation coefficient between $X$ and $Y$. To estimate

the mutual information between two fingers, we used the six-dimensional position measurements from the two sensors at each finger, estimated the covariance matrices from the time series of movement positions and applied Eq. (3).

Prediction of individual finger movements from movements of other fingers: The movement of each individual finger was predicted from the movements of the other fingers. For six-fingered subjects the prediction was carried out with and without the supernumerary finger; the latter to facilitate comparison with the results from five-fingered subjects. The $x/y/z$-positions of the two sensors at each finger constituted the six-dimensional movement vector of each finger. These six components were individually predicted from the 24- or 30-dimensional movement vectors of the remaining four or five fingers. Prediction was done using linear least-squares and nonlinear support vector regression. We used twofold cross-validation with chronological splits of the data to avoid overfitting. The quality of prediction was quantified by computing the coefficient of determination ($R^2$) between predicted and actual movement for each component of the six-dimensional movement vector and then averaging the $R^2$ values across the six dimensions. We used support vector regression with a Gaussian kernel and the hyperparameters (i.e. the kernel width as well as the regularization parameter) were optimized on the training data set. We used the Matlab implementation ("fitrsvm") for support vector regression and optimization of hyperparameters. To reduce computation time the data were downsampled to 120/20 = 6 Hz.

Principal component analysis (PCA) of degrees of freedom[21,28,29]: PCA was performed on the sensor $x/y/z$-positions measured with two sensors at each finger during the object manipulation and the common movement tasks. The cumulated amount of variance captured by an increasing number of principal components is plotted in Fig. 3b and Supplementary Fig. 6B. To compute the effective number of dof we applied two algorithms: the cross-validation PCA with Eigenvector method recommended in ref. [30] and the cross-validation PCA method using expectation maximization for missing values as proposed in ref. [31]. Both methods use a cross-validation procedure where the PCA is first computed from training data and then applied to predict the samples of the test data while training and test data set are mutually exclusive[30,31]. In our case we used tenfold cross-validation and chronologically split the movement data separately for each task into ten parts using in each fold nine of those parts in the training and one part in the test data. The first and last 10 s of the test data set were excluded for each task to avoid any influence of the training on the test data due to the auto-correlation of the movement. The mean squared error between prediction and actual data was computed as a function of the number of principal components. The number of principal components which yielded the smallest error was used as an estimate for the effective number of dof and was computed for each subject separately. For each subject we averaged the determined number of principal components across both methods[30,31] and used this as an estimate of the number of degrees of freedom (Fig. 3c, Supplementary Fig. 6C).

Information theoretic analysis of degrees of freedom: In addition to the PCA analysis described in the previous section, we analyzed the degrees of freedom using information entropy. In contrast to the PCA, the analysis of information entropy takes into account potential nonlinear relationships between finger movements. Information entropy, on the other hand, requires an estimate of the joint probability distribution of the finger movements. To compute this joint probability distribution, we discretized the finger movements by classifying the movement state of each finger into one of three conditions from the set MS = {rest, flexion, extension}, based on the movements of the distal and proximal interphalangeal joints. Spherical coordinates (distance, polar and azimuth angle) of the distal sensor relative to its proximal sensor were computed. PCA was performed on the polar and azimuth angles and the movements along the first principal component were used to represent the movements of each finger. For each finger, the first derivative $v$ of the first PC was calculated as the difference between two consecutive time bins and used to derive the current movement state based on a threshold $\mu = 0.3$ SD($v$): flexion for $v < -\mu$, extension for $v > \mu$, rest otherwise. Different threshold values ($\mu = 0.4$ SD($v$) or $\mu = 0.1$ SD($v$)), as well as different set of states (only two states: flexion for $v < 0$ and extension for $v > 0$), did not change our general conclusion regarding the comparison of the information entropy between five- and six-fingered subjects. We computed the information or Shannon entropy ($H$) of the joint probability distribution of the movement states of all fingers ($p$):

$$H = - \sum_{s_1 \in MS} \sum_{s_2 \in MS} \cdots \sum_{s_n \in MS} P(s_1, s_2, \dots, s_n) \log_2 [(s_1, s_2, \dots, s_n)], \qquad (5)$$

where $s_i \in$ MS is the state of finger $i$. For $n$ fingers the number of different movement states is $3^n$ and the maximum entropy is therefore $\log_2(3)^n$ which is obtained when all possible movement states have equal probability.

Joint movement of thumb, index and supernumerary finger: For each time point we computed the movement speed for each finger as the magnitude of its three-dimensional velocity vector at the fingertip. We then classified the movement state of each finger in each time point as either "rest" or "moving" by comparing the speed to a threshold value which was chosen as the 10th, 30th or 50th percentile of the speed distribution across all time points and all fingers. From these data we estimated the conditional probabilities that thumb and index finger or thumb alone or index finger alone were moving given the supernumerary finger

was moving. These conditional probabilities were estimated for the three speed thresholds (Fig. 3e, Supplementary Fig. 6E).

**Video game for six fingers**. Polydactyly subjects sat in front of a computer monitor (DELL U2713HM) approximately 0.6 m from the screen, on which six target boxes were displayed in the lower centre of a black screen. During the experiment, oscillating cursors passed through the target boxes (Fig. 3g and Supplementary Movie 6). Each of these oscillating squares had a different frequency within a predefined range. The individual target boxes could be "touched" by pressing a corresponding key on a standard computer keyboard. Keys were chosen to match the hand geometry of individual subjects to ensure pressing the keys was comfortable. The subjects were instructed to track the oscillating cursors and to press the corresponding button once the cursor was within its associated target box. If the button was pressed within this time window, it counted as a correct press, if it was pressed outside it was counted as a false press. The number of correct and false presses were summed over all fingers and accumulated over the time of the trial.

The performance of the subjects was rated on their accuracy (correct presses/ target count) and error rate (false presses/all presses). The aim was to increase accuracy while decreasing the error rate. At the beginning of each trial the target accuracy and the error rate threshold was set according to the level (Supplementary Table 1); each level was defined by the movement speed of the oscillating cursors and thresholds on the accuracy and the error rate. Once the subject crossed both thresholds, the participant was expected to maintain their performance above the accuracy and below the error threshold for 2 minutes, at which point the trial would end and the level would be increased. For each subsequent level, the accuracy threshold was set 10% higher and the error rate was set 10% lower. If the subject was able to cross the 70% threshold for accuracy and go below the 30% threshold for the error rate, the oscillation frequency range was increased by 0.05 Hz. After increasing the oscillation frequency, the accuracy threshold and error rate were set back to the original value of 50%. See Supplementary Table 1 that highlights the parameter values associated with different levels. If the subject was not able to reach the next level within 7 minutes, the trial was aborted and after a short break, the subject was asked to repeat the same level.

During each trial, the following additional visual feedback was presented to the subject. If no key was pressed, the target boxes were displayed in white. Pressing a key while no cursor was in the corresponding box, i.e. a false press, the target box turned red. Pressing a key while a cursor was in the corresponding box, i.e. correct press, the target box turned blue. Below the target boxes, two bars gave visual feedback about the subject's overall performance. The upper bar reflected the accuracy and the lower bar the error rate. If the accuracy of the subject increased, the accuracy bar filled up and vice versa. At the same time, decreasing the error results in filling of the error bar, such that an error rate equal to 0 resulted in an entirely filled bar, i.e. the value of 1-error rate was presented. Each bar was red until the subject crossed the set threshold of the corresponding bar, at which point it turned green. The threshold values were shown as gray markers on the bars. As soon as both bars turned green, a red countdown of 120 s appeared in the lower centre of the screen. If one bar turned red again before the time was expired, the countdown was reset to 120 s and disappeared until both bars were green again. Furthermore, each cursor individually appeared in red (if below) or green (if above) for the performance threshold in relation to the individual performance of the corresponding finger, so the subjects had an indication of which finger required improvement.

The evolution of performance is shown in Fig. 3h. Subjects were tested for five consecutive days as well as 10 days after. The subjects performed the task for 1 h per day. The subjects had to use two different finger combinations to press the keys; either all six fingers from the right hand or the right hand but replaced the SF with the index finger of the left hand (Fig. 3h).

**Statistical analysis**. For comparing two independent samples we used the nonparametric, two-sided Wilcoxon ranksum test and computed 95% confidence intervals on the effect size (i.e. the difference of the population means) by using the two-sample pooled t-interval. For comparing two paired samples we used the nonparametric, two-sided Wilcoxon signed rank test and computed 95% confidence intervals on the effect size by using the paired t-interval. All reported confidence intervals reflect the mean for five-fingered subjects subtracted from the mean for six-fingered subjects, i.e. positive values indicate larger values for six-fingered subjects.

To assess the correlation between two variables we computed the Pearson correlation coefficient. We did not assess the statistical significance of the Pearson correlation coefficient as the samples across which correlations were computed were not independent.

**Reporting summary**. Further information on research design is available in the Nature Research Reporting Summary linked to this article.

## Data availability

The data are not available because the anonymity of the participants cannot be guaranteed due to their anatomical peculiarities.

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

## Acknowledgements

We thank the subjects of this study for their availability and flexibility. We thank Tomoki Arichi, Glauco Caurin, Sofia Dall'Orso, Ana Dos Santos Gomes, Jonathan Eden, Alessandro Farnè, Dollyane Muret, Luca Rosalia and Marco Solca for technical assistance. We thank Claudia Clopath, Nicolas Rojas and Stephen Scott for comments on an earlier version of this manuscript and Dmitry Kobak for comments on parts of the analyses. This research was funded in part by the German Research Foundation (DFG) through grants no INST 39/1014-1 and INST 39/963-1, the state of Baden-Württemberg through bwHPC and the Struktur- und Innovationsfonds (SI-BW), the Swiss National Science Foundation (PP00P3_163951 / 1), by EU FP7 grants PEOPLE-ITN-317488-CONTEST, ICT-611626 SYMBITRON, H2020 ICT-644727 COGIMON, Minded Program—Marie Skłodowska-Curie grant agreement no. 754490 and by the grant UK EPSRC MOTION EP/NO29003/1.

## Author contributions

Conceived and designed the study: C.M., L.B., M.M., H.C., A.S., E.B.; Polydactyly subjects recruitment: C.M., L.B., E.B.; Performed the experiments: L.B., M.A., M.M., H.C., M.B., A.S.B., T.P., R.S., O.B.; Analyzed and interpreted the data: C.M., L.B., M.A., M.M., H.C., T.P., A.C., A.S., E.B.; Wrote the manuscript: C.M., M.A., A.S., E.B.; All co-authors have read and edited the manuscript and agree with its content.

## Additional information

**Competing interests:** The authors declare no competing interests.

