## [Peer Review File · Nature Communications]

Reviewers' comments:

Reviewer #1 (Remarks to the Author):

Augmented manipulation ability in humans with six fingered hands
By Mehring et al.

The authors provide morphologic, kinematic and imaging data of 2 participants with six fingers on each hand. In addition, the authors used multi-object manipulations and applied a training involving all 6 fingers usually performed with two hands.

Overall the case reports are highly interesting and the questions asked are important and well selected. In addition methods for testing these questions are appropriate and the visualization and embedding of results in the literature is very good. The article is highly fascinating and the conclusions are well drawn.

Especially the fMRI-part is performed methodologically appropriately.

I suggest to publish this report with minor changes.

Minor:

It has to be considered that only one participant was able to be included in the MRI-task. Therefore this is a one-participant finding! This might be mentioned as a limitation.

Fig 2:

Please indicate dimension for the y and x-axis.

Reviewer #2 (Remarks to the Author):

The authors analyzed the neuromechanics and manipulation abilities of 2 6-fingered individuals. The anatomical analysis, based on MR imaging, revealed that the supernumerary finger (SF) had extra muscles and nerves although some anatomical ambiguity existed which placed the SF in between the long fingers and the thumb. Individuals were able to move the SF independently from other fingers. More interestingly, 6-fingered individuals were able to integrate the SF in coordinated movements with the other fingers which exhibited greater complexity that exceeded the complexity achievable with 5 fingers. This complexity was present in the absence of evidence for any motor deficits.

The authors conclude that the human brain can "deal with the increased complexity to control an extra limb and yield functional advantages from it".

This is an original and thought-provoking study on a relevant topic in motor control. Although all experiments and analyses have been carefully designed and performed (some minor comments see below), I am not sure that the general conclusions are sufficiently based on the findings.

Specific comments to the authors:

1) It remains unclear if conclusions based on findings in polydactyl individuals can be extended to normodactyl individuals. Hexadactyl individuals differ from normodactyl individuals in whom a SF is added by the fact that their SF is present during their individual ontogenesis, allowing for its embodiment and may trigger neural developmental processes which may no longer be possible in the mature brain of normodactyl individuals. Because the brain of a polydactyl individual is likely to differ from the brain of a normodactyl individual by its developmental trajectory, the conclusion that the human brain is able to control a body with additional degrees-of-freedom may not be justified.

2) The questions raised in the Intro ("First, is the movement of the additional finger actuated by other fingers' muscles, or does it have its own dedicated muscles and nerves? Second, how independent is the extra finger from the other fingers? Does its movement accompany the movement of the common finger, like in the little and ring fingers, or does it move independently from other fingers like the thumb?") are misleading. A quick search on the topic reveals a variety

of different phenotypes of polydactyly. Clearly, the questions posed are relevant in relation to the specific objective of the paper because meaningful questions about the neural control of SF require certain anatomical and biomechanical preconditions to be fulfilled.

3) Information theoretic analysis of degrees of freedom: This was done by discretizing joint positions into flexion, rest and extension. It remains unclear how rest was defined. As most interphalangeal joint cannot extend beyond the neutral state the definition of extension depends on defining the rest position which must deviate from the neutral position.

4) The authors state that the SF endows polydactyly subjects with (Discussion) "a higher sensorimotor ability for manipulation than in normal-bodied individuals." The reported experiments have convincingly shown that the polydactyly individuals have superior abilities with one hand. However, is this level of consideration relevant in relation to the movement space? It may be argued that if the cortex represents actions and action goals, rather than nerves or muscles, the successful integration of a SF into the repertoire of movements of one hand has not expanded the capabilities of the polydactyly individual compared to normodactyl subjects as long as individuals using their second hand accomplish the same goal at similar speed and accuracy. Can the authors confidently exclude the possibility that the superior manipulation ability with one hand is accomplished by resources that are normally used to control the other hand? A prediction of this idea would be that simultaneous actions performed by the other hand should be performed at less accuracy in polydactyl subjects than in normodactyl subjects.

The authors' hypothesis of superior motor capabilities of polydactyl individuals would be strengthened if examination of the combined motor capabilities of both hands would reveal superior abilities compared to normodactyl subjects.

Minor:

1) Page numbers are missing.

Reviewer #3 (Remarks to the Author):

The authors analyzed mechanical characteristics, manipulation abilities, and MRI cortical projections of two polydactyly subjects (17 years old and 52 year old, son and mother), who both have six anatomically fully developed fingers on the two hands. They explored finger maximal forces, noise level, mutual finger interdependence, and performance in a computer game. The authors conclude that the 6-fingered subjects could carry out, with one hand, tasks that normally require two hands. They also conclude that a body with significantly more degrees-of-freedom can be controlled by the human nervous system without causing motor deficits or impairments and even provide superior manipulation abilities.

This is a study of broad interest. Overall, the experiments were performed competently. However, the paper is hard to read, and in places it reads as advertising rather than a scientific report. A number of conclusions seem to represent over-statements not supported by the data.

It is not clear what the authors mean under "the range of possible movements increases exponentially with every dof" (page 2). How is "the range" quantified? Are the authors sure that the increase is exponential, not a power function, or some other function? Please, explain or present references.

The authors use broadly the term "enslaving" introduced by Zatsiorsky without references.

Figures 2 and 3 are very hard to understand. Their multiple panels present a lot of information without explaining how it was obtained (what was measured and computed in what kind of experiments and under what kind of instruction). The text and caption are both very brief. Some of the descriptions are too informal, such as "some dependence" or "similar dependences". Overall, there is lack of appropriate statistical support for many of the statements.

The authors conclude that "movements of the 6-fingered hands had increased complexity relative to common 5-fingered hands", which by itself is not very surprising. The video game analysis does not suggest that 6-fingered persons can do tasks typically performed with two hands by 5-fingered subjects. The game was specifically designed to be controlled by 6 effectors, which gives an unfair advantage to 6-fingered persons. It would be fair to compare the performance in 5-object games and 7-object games.

An important issue is whether the two subjects studied by the team represent typical 6-fingered (or other >5-fingered) persons or they are a special rare case. In the experience of the reviewer, most extra fingers are not as functional or may even be detrimental to function compared to the five "regular fingers". Generalization of the conclusions from this study to other cases of polydactyly and to the ability of 5-fingered persons to control devices with multiple DOFs is at this time pure speculation. In particular, the statement "there is value in augmenting normal 5-fingered hands with an artificial supernumerary finger" seems pure science fiction.

Reviewer #4 (Remarks to the Author):

This is a very enjoyable, rigorous, original, and interesting paper with important future implications.

I have a few questions:

1. Why was enslaving measured with maximal voluntary force? This will miss subtle differences between fingers at lower levels of force.
2. The authors state that there were 6 distinct finger representations and that they were associated with an increased volume of the hand representation: "as the total area activated by the movements of the 6 fingers was larger than that activated by the movements of the 5 fingers in 5-fingered individuals." It is notoriously difficult to ascertain true extent of activation with functional imaging as t-maps are NOT extent and cannot be taken to be. What was done here?
3. In figure 2E, I am not sure why there is such overlap between two of the fingers in the 5 fingered control. This is quite different from, for example, Ejaz, Hamada, and Diedrichsen Nature Neuroscience 2015 figure 3D.

We thank the reviewers and the editor for their thorough examination of our manuscript, their comments and interesting suggestions. Please find below a point-by-point description of how we have addressed these comments in the attached revision. Please also note that we have introduced a new Methods section which contains the material that was previously presented in the Supplementary Methods.

Reviewer #1

Augmented manipulation ability in humans with six fingered hands, by Mehring et al.

The authors provide morphologic, kinematic and imaging data of 2 participants with six fingers on each hand. In addition, the authors used multi-object manipulations and applied a training involving all 6 fingers usually performed with two hands.

Overall the case reports are highly interesting and the questions asked are important and well selected. In addition methods for testing these questions are appropriate and the visualization and embedding of results in the literature is very good. The article is highly fascinating and the conclusions are well drawn.

Especially the fMRI-part is performed methodologically appropriately.

I suggest to publish this report with minor changes.

Minor:

It has to be considered that only one participant was able to be included in the MRI-task. Therefore this is a one-participant finding! This might be mentioned as a limitation.

Thank you for this comment; we have added this information to the main text (at lines 132, 138-140).

Fig 2: Please indicate dimension for the y and x-axis.

Fig. 2E is the result of classical multi-dimensional scaling (MDS) which computes a low-dimensional map of the data preserving the pairwise distances in the original high-dimensional dataset. In our case the original high-dimensional dataset was comprised of fMRI voxel activations for each finger. As distance measure we used the Mahalanobis distance (as was also used in a similar analysis by Ejaz et al. 2015, Nature Neuroscience, reference 14 in the manuscript). This distance measure is unitless and therefore the 2D projection obtained by MDS is unitless as well. Please note that in the original submission we used the distance measure “1-Pearson correlation coefficient” which is unitless as well.

In addition, there is no ‘dimension’ that could be indicated on the axes besides calling them “First MDS coordinate” and “Second MDS coordinate”. Moreover, MDS induces an arbitrary rotation and hence any rotated map is as valid as the actual map shown in Fig.2E. We, therefore, do not think it is helpful to label the axes and instead show only the scaling.

Reviewer #2

The authors analyzed the neuromechanics and manipulation abilities of 2 6-fingered individuals. The anatomical analysis, based on MR imaging, revealed that the supernumerary finger (SF) had extra muscles and nerves although some anatomical ambiguity existed which placed the SF in between the long fingers and the thumb. Individuals were able to move the SF independently from other fingers. More interestingly, 6-fingered individuals were able to integrate the SF in coordinated movements with the other fingers which exhibited greater complexity that exceeded the complexity achievable with 5 fingers. This complexity was present in the absence of evidence for any motor deficits.

The authors conclude that the human brain can “deal with the increased complexity to control an extra limb and yield functional advantages from it”.

This is an original and thought-provoking study on a relevant topic in motor control. Although all experiments and analyses have been carefully designed and performed (some minor comments see below), I am not sure that the general conclusions are sufficiently based on the findings.

Specific comments to the authors:

1) It remains unclear if conclusions based on findings in polydactyl individuals can be extended to normodactyl individuals. Hexadactyl individuals differ from normodactyl individuals in whom a SF is added by the fact that their SF is present during their individual ontogenesis, allowing for its embodiment and may trigger neural developmental processes which may no longer be possible in the mature brain of normodactyl individuals. Because the brain of a polydactyl individual is likely to differ from the brain of a normodactyl individual by its developmental trajectory, the conclusion that the human brain is able to control a body with additional degrees-of-freedom may not be justified.

We agree with the reviewer that the development of motor function and embodiment in hexadactyl subjects may be different from those in normodactyl subjects. While our findings are therefore no proof that the human brain of normodactyl subjects can control additional dof, our results do demonstrate that a human brain (as developed in our polydactyl subjects) is able to control additional dof. We added a corresponding discussion to the manuscript at lines 260-264.

2) The questions raised in the Intro (“First, is the movement of the additional finger actuated by other fingers’ muscles, or does it have its own dedicated muscles and nerves? Second, how independent is the extra finger from the other fingers? Does its movement accompany the movement of the common finger, like in the little and ring fingers, or does it move independently from other fingers like the thumb?”) are misleading. A quick search on the topic reveals a variety of different phenotypes of polydactyly. Clearly, the questions posed are relevant in relation to the specific objective of the paper because meaningful questions about the neural control of SF require certain anatomical and biomechanical preconditions to be fulfilled.

The reviewer is correct that there are different types of polydactyly in humans, our subjects belonging to the preaxial subgroup with the supernumerary finger between thumb and index finger (while there is also a postaxial group and a central group). We agree with the reviewer that the anatomical and biomechanical properties of the hand will depend on type of polydactyly, and have thus added these important points in various places in the manuscript (lines 84-86, 230-231, 237, 246-247). While the answers to our questions will depend on the type of polydactyly (and possibly on the

individual), the questions raised in the manuscript should be useful to characterise the anatomy and functionality of any type of polydactyly.

3) Information theoretic analysis of degrees of freedom: This was done by discretizing joint positions into flexion, rest and extension. It remains unclear how rest was defined. As most interphalangeal joint cannot extend beyond the neutral state the definition of extension depends on defining the rest position which must deviate from the neutral position.

This seems to be a misunderstanding, as discretisation of the movement was based on movement velocity (not position). Hence, rest was defined as movement speed below a certain threshold while extension was defined as movement in one direction with a speed above threshold and likewise flexion as movement in the opposite direction. Details are explained in the Methods (lines 514-523).

4) The authors state that the SF endows polydactyly subjects with (Discussion) “a higher sensorimotor ability for manipulation than in normal-bodied individuals.” The reported experiments have convincingly shown that the polydactyly individuals have superior abilities with one hand. However, is this level of consideration relevant in relation to the movement space? It may be argued that if the cortex represents actions and action goals, rather than nerves or muscles, the successful integration of a SF into the repertoire of movements of one hand has not expanded the capabilities of the polydactyly individual compared to normodactyl subjects as long as individuals using their second hand accomplish the same goal at similar speed and accuracy. Can the authors confidently exclude the possibility that the superior manipulation ability with one hand is accomplished by resources that are normally used to control the other hand? A prediction of this idea would be that simultaneous actions performed by the other hand should be performed at less accuracy in polydactyl subjects than in normodactyl subjects.

The authors’ hypothesis of superior motor capabilities of polydactyl individuals would be strengthened if examination of the combined motor capabilities of both hands would reveal superior abilities compared to normodactyl subjects.

We thank the reviewer for raising this interesting issue. What our results show is that the observed polydactyly subjects have “a higher sensorimotor ability for manipulation *with one hand* than normal-bodied individuals”, as is now clearly indicated in the revised text at lines 59 and 243-246. We developed experiments to analyse the manipulation in one hand because many tasks in everyday life (such as writing, tool use, haptic exploration, small object manipulation, mobile phone use) require complex control in only one hand. We do not pretend that our polydactyly subjects have superior abilities in manipulation tasks involving the two hands (or in tasks with other limbs as well such as playing a Bach fugue on the organ using both hands and feet). However, having hands with six well-controlled fingers provide our polydactyly subjects richer motion behaviours (with one hand) than is possible with a normal 5-fingered hand, as well as the ability to carry out with one hand operations that require people with normal 5-fingered hands using their two hands, as our results demonstrate.

Minor: Page numbers are missing.

We have added line and page numbers to the revised manuscript.

Reviewer #3

The authors analyzed mechanical characteristics, manipulation abilities, and MRI cortical projections of two polydactyly subjects (17 years old and 52 years old, son and mother), who both have six anatomically fully developed fingers on the two hands. They explored finger maximal forces, noise level, mutual finger interdependence, and performance in a computer game. The authors conclude that the 6-fingered subjects could carry out, with one hand, tasks that normally require two hands. They also conclude that a body with significantly more degrees-of-freedom can be controlled by the human nervous system without causing motor deficits or impairments and even provide superior manipulation abilities.

This is a study of broad interest. Overall, the experiments were performed competently. However, the paper is hard to read, and in places it reads as advertising rather than a scientific report. A number of conclusions seem to represent over-statements not supported by the data.

It is not clear what the authors mean under "the range of possible movements increases exponentially with every dof" (page 2). How is "the range" quantified? Are the authors sure that the increase is exponential, not a power function, or some other function? Please, explain or present references.

The following example may help to clarify what we mean: assume each finger can be in one of the three different movement states {rest, flexion, extension}. Then, if f fingers can be moved independently, there are 3^f different movement states. Hence, there is an exponential increase of possible movement states with the number of fingers. A similar argument could be applied to the number of joints or muscles. Likewise, in continuous space the volume of a mathematical space grows exponentially with every additional dimension. We have added a note in the Supplementary material with this explanation.

The authors use broadly the term "enslaving" introduced by Zatsiorsky without references.

Thank you for pointing this out. We are now citing the paper by Zatsiorsky, Li and Latash (2000) on "Enslaving effects in multi-finger force production" as reference (18).

Figures 2 and 3 are very hard to understand. Their multiple panels present a lot of information without explaining how it was obtained (what was measured and computed in what kind of experiments and under what kind of instruction). The text and caption are both very brief. Some of the descriptions are too informal, such as "some dependence" or "similar dependences".

Thank you for this comment. Looking back at the submitted manuscript, we noticed that some figures needed further information and descriptions in the captions. As figure captions have a word limit in Nature Communications, we have thus moved the results of the statistical analyses from the captions to the main text and then improved the description in the figure captions as well as the figures itself. We have further improved the main text which refers to those figures. Finally, we have rechecked our Methods section and improved it where appropriate.

Overall, there is lack of appropriate statistical support for many of the statements.

Thank you for this comment. We have previously reported several statistical results in the figure captions. We have now added additional statistical analyses (statistical

tests and confidence intervals) and all statistical results are now reported in the text (rather than in the figure captions) except for the results shown in the new Supplementary Figure 2.

Statistical results are reported throughout the results section (at lines 104-106, 107-108, 122, 124-126, 151-152, 162-165, 167-169, 173-175, 181-186, 188-190, 199-202, 220, caption of Supplementary Figure 2); further details on the statistical methods are reported in the methods section (at lines 601-611).

The authors conclude that “movements of the 6-fingered hands had increased complexity relative to common 5-fingered hands”, which by itself is not very surprising.

We appreciate the reviewer’s opinion, however the increased hand movement’s complexity in our 6-fingered subjects is not trivial, for the following reasons: Firstly, while the 6-fingered hands of our subjects have more joints than a normal 5-fingered hand, this does not necessarily mean that they have additional resources to control the additional fingers. For instance, the muscles of the normal 5-fingered hand may be shared with the sixth finger. In such a situation, the sixth finger may not be able to perform independent movements and the movement complexity of the 6-finger hand is not higher than the 5-finger hand. In the extreme case, one actuator would be sufficient to control the grasping in a hand (with 5 or 6 fingers), where all fingers and joints close or open in a synchronised way. However, the 6-fingered hand of P1 indeed has additional muscles, nerves and motor cortical resources dedicated to moving the extra finger (as appears in Fig.1 and Fig. 2E). Even with these additional resources, it remains unclear whether polydactyly subjects could use these additional muscles and nerves to actuate the SF independently of the other fingers. The results of Fig. 2 show that this is indeed the case. Finally, it was unclear whether our subjects effectively use these additional capabilities in voluntary movements (which was demonstrated by the higher movement complexity relative to 5-fingered hands in Fig.3).

The video game analysis does not suggest that 6-fingered persons can do tasks typically performed with two hands by 5-fingered subjects. The game was specifically designed to be controlled by 6 effectors, which gives an unfair advantage to 6-fingered persons. It would be fair to compare the performance in 5-object games and 7-object games.

The rationale behind the video game is to investigate a task which cannot be performed by 5 fingers in order to test whether the 6th finger is detrimental to the movement of the other fingers or whether it can be used to perform a more complex task. The video game shows that 6-fingered subjects obtained the same performance with 6-fingers from one hand as with the regular 5-fingers from the same hand and the index finger from the other hand; the latter being the way how 5-fingered subjects would typically perform the 6 dof video game. Therefore, we conclude that “this demonstrates the augmented abilities for manipulation enabled by the 6-fingered hand compared to a common 5-fingered hand”. We suggest the possibility of 6-fingered subjects performing with one hand tasks that require individuals with 5-fingered hands to use their two hands. We do not pretend that this is the case for all two hand tasks that 5-fingered individuals can perform.

An important issue is whether the two subjects studied by the team represent typical 6-fingered (or other >5-fingered) persons or they are a special rare case. In the experience of the reviewer, most extra fingers are not as functional or may even be

detrimental to function compared to the five "regular fingers". Generalization of the conclusions from this study to other cases of polydactyly...

We agree with the reviewer that in many polydactyly cases the extra fingers are not functional or may even be detrimental to function which is why we wrote in the introduction (at lines 65-67): "However, supernumerary fingers are often removed at birth (11) as they are deemed not useful and are often not fully developed." We now also mention this important point in the Discussion of the revised manuscript and clearly state that our 6-fingers participants represent one case of polydactyly, i.e. the *preaxial* case of polydactyly with an SF between thumb and index finger (see lines 230-231, 246-247).

....and to the ability of 5-fingered persons to control devices with multiple DOFs is at this time pure speculation. In particular, the statement "there is value in augmenting normal 5-fingered hands with an artificial supernumerary finger" seems pure science fiction.

For several years, roboticists have been attempting to develop extra limbs to augment human movement abilities as cited e.g. in references (1) and (2) of our manuscript. A recent study in *Science Robotics* 3(20) (doi:10.1126/scirobotics.aat1228, ref 6 in the manuscript) even claims to have achieved the control of a third arm by a brain-computer interface. While intuitive and efficient realtime control of additional dofs has indeed not yet been demonstrated (and we are sceptical about some of the developments), supernumerary limb control thus appears as recent developments in science and technology rather than as science fiction. We have extended the Discussion to clarify this point, see lines 260-271. We do not pretend that current robotic supernumerary limbs can augment motor function in humans. However, we find important to confront the robotic developments towards supernumerary limbs. Furthermore, it is plausible that this report of the neuromechanics and functionality in polydactyly will stimulate and may inform future robotic developments.

Reviewer #4

This is a very enjoyable, rigorous, original, and interesting paper with important future implications. I have a few questions:

1. Why was enslaving measured with maximal voluntary force? This will miss subtle differences between fingers at lower levels of force.

We selected the data with maximal forces (MF) in the initial submission, but had recorded similar data requiring force control at 10%, 20% and 30% of MF. These data show similar enslaving between the fingers as the MF data, in particular for 20% and 30% of MF. These new analyses and results are now presented as a new Supplementary Figure 2 and at lines 126-128 as well as lines 327-329 in the revised manuscript.

2. The authors state that there were 6 distinct finger representations and that they were associated with an increased volume of the hand representation: “as the total area activated by the movements of the 6 fingers was larger than that activated by the movements of the 5 fingers in 5-fingered individuals.” It is notoriously difficult to ascertain true extent of activation with functional imaging as t-maps are NOT extent and cannot be taken to be. What was done here?

We thank the reviewer for raising this point. We estimated the extent of the hand area as the number of suprathreshold voxels (F-contrast, $p < 0.05$ FWE) within the considered volumes (M1: BAs 4a and 4p; S1: BAs 3a, 3b, 1 and 2). The F-contrast was designed to identify voxels associated with the movement of at least one finger. We agree with the reviewer that this estimation of hand area is questionable as it does not consider the statistical density function. As this statement further is a minor point of our analysis, we prefer to delete it in the new version (see lines 140-143). We also removed the corresponding sentences in the methods section concerning the statistical comparison between P1 and controls (lines 369-372).

3. In figure 2E, I am not sure why there is such overlap between two of the fingers in the 5 fingered control. This is quite different from, for example, Ejaz, Hamada, and Diedrichsen Nature Neuroscience 2015 figure 3D.

This interesting point prompted us to reanalyse our data. As dissimilarity measure Ejaz et al. 2015 used the cross-validated Mahalanobis distance (cv-MD) while we originally used 1-Pearson correlation coefficient (1-corr). We have now re-analysed our data using the cv-MD (for details see revised Methods section, lines 383-394) and using this dissimilarity measure we obtain much less overlap between the fingers in the 5-fingered control and overall our 5-finger results are now similar to Ejaz et al. (cf. the new Fig. 2E in the revised manuscript).

Besides theoretical reasons for using the cv-MD instead of 1-corr, we also found an increase of the inter-subject reliability, which was 0.62 ± 0.24 for cv-MD versus 0.32 ± 0.14 for 1-corr. We, therefore, decided to use cv-MD instead of 1-corr in the entire revised manuscript, i.e for 6- and 5-finger subjects. The results for the 6-finger subject remains qualitatively similar to the previous results using 1-corr (see Fig. 2E and Supplementary Figure 4).

Despite the different distance measures, a second less important reason for the difference between our results for 5-fingered subjects and the results of Ejaz et al. 2015 was that in the originally submission we plotted 50% confidence ellipses while Ejaz et al. 2015 show s.e.m. ellipses. While s.e.m. ellipses are obtained by dividing the covariance across subjects by the number of subjects, 50% confidence ellipses

are larger by a factor of about 1.18 (the inflation factor was obtained as the square root of the 50% percentile of the chi² distribution with two degrees of freedom). To be more consistent with the existing literature, namely Ejaz et al. 2015, we now plot s.e.m. ellipses. Please note that we also applied a Monte-Carlo based correction for the Procrustes alignment as was done in Ejaz et al. 2015.

We have revised Fig. 2E and Supplementary Fig. 4 to show the new results as well as the Methods section to explain the new analyses methods.

Reviewers' comments:

Reviewer #1 (Remarks to the Author):

The issues have been addressed sufficiently and I have no more comments.

Reviewer #2 (Remarks to the Author):

The authors have responded to the issues raised by the reviewers but do not appear to have provided satisfactory answers to some of the questions. Additionally, the revision in the manuscript does not fully reflect the insight provided in the authors' response to the reviewer's concerns.

1) They state in the rebuttal: "our results do demonstrate that a human brain (as developed in our polydactyl subjects) is able to control additional dof"

However, in the manuscript they write (ln 262 sequ):

"the exceptional manipulation abilities in our polydactyl subjects suggest that it is in principle possible, and may be of value, to augment normal 5-fingered hands with an artificial supernumerary finger"

This statement clearly significantly extends that provided in the rebuttal letter. I disagree with the authors that they have provided evidence to suggest that adding a supernumerary finger to normal 5-fingered hands will provide the recipient of that finger with superior motor abilities. A similar concern has been raised by reviewer #3.

2) If the authors write that (ln. 70) "the neuromechanics and functionality of polydactyl hands raise many questions that have never been investigated" the reader expects findings that generalize to all individuals with a supernumerary finger. Because there are many phenotypes of polydactyly a study on 2 subjects cannot answer the question, for instance, whether in polydactyl subjects the movement of the additional finger is actuated by other fingers' muscles, or may have its own dedicated muscles and nerves. Therefore, I maintain that the first questions raised in the intro are misplaced. The authors needed to address these questions as a prerequisite to meaningfully address the issue of motor control in hexadactyl individuals at a behavioral and physiological level. A similar concern has been raised by reviewer # 3.

3) Superior ability for manipulation. The authors now appropriately limit their conclusions to stating that polydactyl subjects have ". Unfortunately, because this superior ability is demonstrated in one hand only, the study does not inform about the question whether adding additional effectors to 1 hand has expanded motor control abilities in general. Without some consideration of the general motor control ability, the study provides a nice phenomenological demonstration that a sixth finger on 1 hand can, in some people, be functional and useful. However, its value on theories of motor control appears to be limited if we accept that a network interconnected motor areas on both hemispheres is active in controlling hand movements of either side.

Demonstration of uncompromised motor ability in 1 hand, when challenging the other with a task that requires 2 hands in normal 5-fingered subjects, does appear as a feasible control experiment.

Reviewer #3 (Remarks to the Author):

I am satisfied with the authors' response to all the major comments.

Reviewer #4 (Remarks to the Author):

I am happy with the changes made.

Point-by-point reply:

Your revised manuscript entitled "Augmented manipulation ability in humans with six fingered hands" has now been seen again by 4 referees at Nature Communications.

You will see from their comments below that Reviewers #1, #3 and #4 are now happy to recommend publication of your manuscript. However, Reviewer #2 still expresses some concerns.

Having considered Reviewer #2's comments, we have taken the editorial decision that we will not ask you to carry out data collection as Reviewer #2 recommends (e.g. "Demonstration of uncompromised motor ability in 1 hand, when challenging the other with a task that requires 2 hands in normal 5-fingered subjects, does appear as a feasible control experiment.")

While we agree with Reviewer #2 that this additional experiment would be interesting, we do not consider it necessary. However, we do ask you to revise the text of your manuscript to clearly acknowledge the points raised by Reviewer #2 and to avoid making claims that go beyond the data.

We therefore invite you to revise and resubmit your manuscript, taking into account the points raised.

We thank the editors for their careful considerations of the reviewers' feedback. We note the editorial decision to not request the additional experiment suggested by Reviewer 2's third comment.

Reviewer #2

The authors have responded to the issues raised by the reviewers but do not appear to have provided satisfactory answers to some of the questions. Additionally, the revision in the manuscript does not fully reflect the insight provided in the authors' response to the reviewer's concerns.

1) They state in the rebuttal: "our result do demonstrate that a human brain (as developed in our polydactyl subjects) is able to control additional dof"

However, in the manuscript they write (In 262 sequ):

"the exceptional manipulation abilities in our polydactyly subjects suggest that it is in principle possible, and may be of value, to augment normal 5-fingered hands with an artificial supernumerary finger"

This statement clearly significantly extends that provided in the rebuttal letter. I disagree with the authors that they have provided evidence to suggest that adding a supernumerary finger to normal 5-fingered hands will provide the recipient of that finger with superior motor abilities. A similar concern has been raised by reviewer #3.

To avoid suggesting that the results of our study demonstrate that it is possible to obtain similar control abilities with a supernumerary artificial finger, we have modified this sentence to: "The exceptional manipulation abilities in our polydactyly subjects suggest that it may be of value to augment normal 5-

fingered hands with an artificial supernumerary finger.”. See lines 241-243 in the revised manuscript.

2) If the authors write that (ln. 70) “the neuromechanics and functionality of polydactyl hands raise many questions that have never been investigated” the reader expects findings that generalize to all individuals with a supernumerary finger. Because there are many phenotypes of polydactyly a study on 2 subjects cannot answer the question, for instance, whether in polydactyl subjects the movement of the additional finger is actuated by other fingers’ muscles, or may have its own dedicated muscles and nerves. Therefore, I maintain that the first questions raised in the intro are misplaced. The authors needed to address these questions as a prerequisite to meaningfully address the issue of motor control in hexadactyl individuals at a behavioral and physiological level. A similar concern has been raised by reviewer # 3.

The results reported in our manuscript demonstrate the possibility that in polydactyly subjects additional fingers are actuated by dedicated muscles and neural resources and hence, offer augmented manipulation abilities.

To avoid any suggestion that our study addresses the questions in the 3rd paragraph of the Introduction for all kinds of polydactyly, we have now added the following sentence to the end of this paragraph (lines 79-81): “The present case study examines these questions on two subjects with *preaxial* polydactyly with an SF between thumb and index finger.”. Consequently, we have removed the information on the type of polydactyly from the beginning of the result section (lines 85-87).

The number of subjects has already been mentioned in the introduction at lines 59-60, i.e. before the questions.

3) Superior ability for manipulation. The authors now appropriately limit their conclusions to stating that polydactyly subjects have “. Unfortunately, because this superior ability is demonstrated in one hand only, the study does not inform about the question whether adding additional effectors to 1 hand has expanded motor control abilities in general. Without some consideration of the general motor control ability, the study provides a nice phenomenological demonstration that a sixth finger on 1 hand can, in some people, be functional and useful. However, its value on theories of motor control appears to be limited if we accept that a network interconnected motor areas on both hemispheres is active in controlling hand movements of either side. Demonstration of uncompromised motor ability in 1 hand, when challenging the other with a task that requires 2 hands in normal 5-fingered subjects, does appear as a feasible control experiment.

As stated in the paper at lines 60 and 233, our study is limited to the motor ability with one hand, corresponding to many activities of daily living. Though interesting, the additional experiment suggested by this reviewer would point

to a different question and may actually lead to a series of further experiments as the motor abilities with the human bodies are not limited to the second hand but may include e.g. the feet or the head movements.